# Thickness Effects on Boron Doping and Electrochemical Properties of Boron-Doped Diamond Film

**DOI:** 10.3390/molecules28062829

**Published:** 2023-03-21

**Authors:** Hangyu Long, Huawen Hu, Kui Wen, Xuezhang Liu, Shuang Liu, Quan Zhang, Ting Chen

**Affiliations:** 1School of Materials Science and Hydrogen Energy, Foshan University, Foshan 528000, China; 2Guangdong Key Laboratory for Hydrogen Energy Technologies, Foshan 528000, China; 3National Engineering Laboratory for Modem Materials Surface Engineering Technology, Institute of New Materials, Guangdong Academy of Sciences, Guangzhou 510651, China; 4School of Materials and Mechanical Engineering, Jiangxi Science and Technology Normal University, Nanchang 330013, China; 5Liaohe Petroleum Exploration Bureau Limited Material Branch, Panjin 124010, China

**Keywords:** diamond, chemical vapor deposition, film thickness, boron doping, electrochemistry

## Abstract

As a significant parameter in tuning the structure and performance of the boron-doped diamond (BDD), the thickness was focused on the mediation of the boron doping level and electrochemical properties. BDD films with different thicknesses were deposited on silicon wafers by the hot filament chemical vapor deposition (HFCVD) method. The surface morphology and composition of the BDD films were characterized by SEM and Raman, respectively. It was found that an increase in the BDD film thickness resulted in larger grain size, a reduced grain boundary, and a higher boron doping level. The electrochemical performance of the electrode equipped with the BDD film was characterized by cyclic voltammetry (CV) and electrochemical impedance spectroscopy (EIS) in potassium ferricyanide. The results revealed that the thicker films exhibited a smaller peak potential difference, a lower charge transfer resistance, and a higher electron transfer rate. It was believed that the BDD film thickness-driven improvements of boron doping and electrochemical properties were mainly due to the columnar growth mode of CVD polycrystalline diamond film, which led to larger grain size and a lower grain boundary density with increasing film thickness.

## 1. Introduction

Diamond is a unique insulated material with the highest atomic density and hardness. However, it can be transformed into a conductive diamond by incorporating boron with an electron-deficient structure into the diamond lattice, which makes the diamond an electronic acceptor. The obtained conductive boron-doped diamond (BDD) film has been endowed with excellent characteristics, thus opening up a new avenue for the research of diamond in the fields of electrochemical analysis [1], electrochemical degradation [2], bioanalysis [3], superconductivity [4], and field emission [5]. As an electrode material, BDD-based materials have many advantages over traditional electrode materials, such as metal, glass carbon, and pyrolytic graphite [6]. This is because BDD has a wide potential window, a very low background current, a high surface stability, and a high anti-pollution performance in electrochemical applications [7,8]. In addition, BDD is an ideal material for biosensing and constructing bio-interface platforms due to its biological inertness and easily regulated surface termination for the further modification of biological molecules [9,10].

Although the performance of BDD films is dictated by several factors, such as boron doping level, surface morphology, non-diamond carbon content, and surface terminal, the boron doping level plays a critical role in enhancing BDD performance [7,11,12,13]. Many works reveal that the intrinsic conductivity of BDD films can be significantly improved by boron doping at a certain level, which is because the boron atoms incorporated in the diamond lattice can be used as charge acceptor to produce effective hole doping and, in turn, to increase the number of carriers in the BDD film. As the boron doping concentration increases, the BDD film can be converted from an insulator to an electrical conductor [14]. Metal-like and semiconducting properties can also be generated in the case of [B] > 10^20^ and [B] < 10^19^ cm^−3^, respectively [15]. Apart from electrical conductivity, surface electrochemistry also be affected by boron doping. A higher boron doping level has been reported to result in a larger active area and a faster heterogeneous electron transfer rate [16,17]. This effect also extends to the surface conductivity of BDD film. It is well known that the hydrogen termination on the BDD surface increases surface conductivity, while the oxygen termination reduces surface conductivity. However, the surface conductivity of BDD with excessive boron incorporation cannot be significantly changed, even if the hydrogen-terminated surface is converted into an oxygen-terminated one [18,19]. Not only that, BDD thin-film electrodes with high boron doping levels are easier to convert from oxygen to hydrogen termination by cathodic polarization [20]. In addition, the results of surface grafting aryldiazonium salt on BDD films showed that the increase of boron doping level on the surface could improve the surface grafting efficiency [9]. Boron doping also affects the mechanical properties of BDD films. Friction-wear experiments were performed on BDD films with different doping levels, and it was found that the wear rate increased linearly with the increase of the boron doping level [21]. As for the etching of the BDD film in the presence of high-temperature steams, it has been proved that the higher the boron doping level, the higher the specific surface area and capacitance obtained by etching [22]. Of course, in some cases, a lower boron doping level is more conducive to the occurrence of electrochemical reactions on the BDD surface. Studies have shown that a BDD electrode with a low boron doping level can depress the oxidation of hydrogen peroxide during hydrolysis and promote the formation of hydrogen peroxide and the degradation of formic acid [23]. In the albumin degradation process, the low defective surface of BDD electrodes caused by a lower doping level can also reduce electrode surface contamination and improve its stability [24]. Therefore, it is essential to control the incorporation of boron into the diamond film.

For polycrystalline diamonds, the boron doped into the BDD film is influenced by many factors. First, the content of boron in the feed gas directly affects the doping in the film. However, for some films with high doping content, an increase in the boron content in the feed gas reduces the grain size, thus introducing more sp^2^ phases and defects [25]. These sp^2^ phases and defects are easy to be transformed or corroded in the electrochemical process, which reduces the anti-pollution ability, analytical stability, and life of the electrode. Certainly, a critical boron doping level exists for the production of these defects. This value depends on the CH_4_/H_2_ ratio and the BDD growth direction [26]. Second, the BDD film microstructure exerts an impact on the boron content of the film. Numerous studies have shown the uneven incorporation of boron into a diamond. Ushizawa et al. [27] found that the (111) facets are more favorable (nearly ten times) for boron incorporation than the (100) facets. Barnard et al. [28] believed that boron is likely to be positioned at the grain boundary of thin-film samples during the doping process. However, other researchers concluded that the boron was not enriched in grain boundary [29]. The controversy over the incorporation of grain boundaries may be due to the differences in research methods and processes. Lu et al. [30] proposed that boron not only exists in the diamond lattice but also can be enriched in the twin boundaries and defect centers inside the diamond grain.

Therefore, precise control of the incorporation of boron in the BDD film is still a challenge. Generally, the increase of CVD polycrystalline diamond film thickness greatly changes the surface structure, morphology, and composition. This series of changes in microstructure characteristics impact the boron doping of the film, which, in turn, influences the electrochemical performance of the BDD electrode. Therefore, it is significant to systematically study the effect of the BDD film thickness on boron content and electrochemical performance.

In this work, the BDD film thickness was finely tuned by adjusting the deposition time during the HFCVD process. The controllable thickness variation is thus conducive to the subsequent investigation of the impact of the BDD film thickness on the boron doping extents and electrochemical performance. The surface morphology and composition of the BDD films were characterized by SEM and Raman spectroscopy, respectively. The basic electrochemical properties of the BDD films were examined by cyclic voltammetry (CV) and electrochemical impedance spectra (EIS).

## 2. Results and Discussions

Figure 1 presents the surface morphologies of different BDD films under SEM. It can be seen that the grain size of BDD films evidently increases with the deposition time from 1 to 4 h. The grain size of the 0.1B-1 and 0.4B-1 samples are measured to be ~500 nm, much smaller than those of 0.1B-4 and 0.4B-4 prepared by 4-h deposition (~5000 nm). Additionally, 0.1B-4 and 0.4B-4 possess faceted grain and clear grain boundaries. For the films with the same deposition time, the grain size slightly decreases as the additional amount of the boron source increases. In addition, it can be seen from the cross-section views that the thickness of the films is about 0.7 µm and 4 µm at the deposition duration of 1 h and 4 h, respectively. Furthermore, the thickness of the films deposited under 0.4 sccm is slightly thinner than that of 0.1 sccm can be confirmed. That is likely because the doping of more boron atoms into the diamond makes it more likely to gather boron atoms at the grain boundaries, inducing the formation of more structural defects. Consequently, the nucleation rate can be increased, thus hindering grain growth and refining the grains [31,32].

Figure 2 shows the Raman spectra of different BDD films. The Raman spectra present the characteristic asymmetric peak at 1332 cm^−1^, characteristic of a diamond, of which the intensity is reduced after doping. This asymmetry is derived from the Fano effect, caused by the quantum interferences between center optical phonon and continuous electronic states in a covalent semiconductor. Generally, an increase in the boron doping level drives the peak at 1332 cm^−1^ to be more asymmetric [33]. In addition, two broad peaks at 500 and 1200 cm^−1^ can be directly related to boron doping. The wavenumber shifts of the peak at about 500 cm^−1^ are associated with the boron doping level: the higher boron doping level results in a red-shift of the peak to a greater extent, with increased intensity. In contrast, the peak at 1200 cm^−1^ does not shift [34]. The Raman bands at 1350 and 1580 cm^−1^ represent graphite D and G peaks, corresponding to the sp^2^ vibration of disordered carbon and sp^2^ vibration of a complete graphitic lattice, respectively [35].

On the other hand, the characteristic bands of non-diamond species at about 1350 and 1500 cm^−1^ cannot be detected, indicating the good quality of the film. Regardless of the boron flow (0.1 or 0.4 sccm), the Raman signal of silicon (a sharp peak of approximately 520 cm^−1^) can be observed for the BDD films with 1-h deposition. In contrast, such a signal cannot be found for the BDD films with 4 h deposition. This is mainly due to the films with 1 h deposition being so thin that the Raman laser can penetrate the BDD films to collect the information of the substrate silicon. The tenuous existence of silicon signals is unfavorable for the comparative analysis of the peaks at 500 cm^−1^ as a result of boron incorporation. Nevertheless, it is still possible to discern the difference in the peak intensity (enhanced) and position (red-shifted), as generated by an increase in the film thickness. Such a change is more striking for the films prepared with a boron flow of 0.1. Concerning the samples with the deposition time increasing to 4 h, a significant increase in the intensities of the Raman peaks at 500 and 1200 cm^−1^ can be noted. Meanwhile, the intensity of the peaks at 1332 cm^−1^ becomes lowered together with enhanced asymmetry with prolonged deposition. The existence of this asymmetry indicates that the boron doping level exceeds 2 × 10^20^ cm^−3^, thus being considered metal-like [36,37]. All these evolutions of characteristic peaks reveal that the increase of the BDD film thickness can elevate the boron doping level. In addition, the comparison of the BDD films prepared at different boron flow rates in the feed gas but with the same deposition time (e.g., 0.1B-1 vs. 0.4B-1, and 0.1B-4 vs. 0.4B-4) presents that the boron flow rate increase leads to a declined intensity of the peak at 1332 cm^−1^, thus revealing that the boron incorporation into the films can be promoted by increasing boron flow rate.

The microstructure and composition of the BDD film surface significantly affect the electrochemical performance. The properties of the four electrodes were tested in a K_3_[Fe(CN)_6_] solution by CV. Figure 3 presents a CV diagram for different electrodes in a mixed solution of 1 mM K_3_[Fe(CN)_6_] and 0.1 M KCl at a scan rate of 100 mv/s. A very large peak separation (ΔE_p_) of the samples with 1-h deposition can be noted, which can be calculated to be 300 (±3) and 270 (±3) mV for 0.1B-1 and 0.4B-1, respectively. A prolonging deposition to 4 h lowers the ΔE_p_ values to 80 (±2) and 81 (±2) mV measured for 0.1B-4 and 0.4B-4, respectively. At the same time, the peak current is notably enlarged. These results indicate that the surface state and electrochemical performance of the electrode can be markedly changed by increasing the film thickness. The notable reduction of ΔE_p_ with increasing film thickness indicates that the electron transfer rate and reversibility become higher and better, respectively. The value of 80 mV is slightly larger than the theoretical value of ΔE_p_ (i.e., 59 mV) for the single electron transfer reversible system. The slight sacrifice of the reversibility may be derived from the presence of more boron impurity centers, which cause increased scattering in the diamond [16].

EIS is applied to analyze the kinetics of the electrochemical reaction occurring on the electrode, which can reflect the characteristics of the electrode structure and electron transfer behavior. The typical Nyquist plot includes a semicircle part in the higher-frequency region and a linear part in the low-frequency region, corresponding to the electron transfer rate control and diffusion control processes, respectively. Figure 4 shows the EIS of different electrodes in a mixed solution of 1 mM K_3_[Fe(CN)_6_] and 0.1 M KCl at the open circuit potential over a frequency range of 0.01 to 10^6^ Hz. It can be observed that two semicircles at the high-frequency region and a linear line at the low-frequency region can be noted for the four BDD electrodes. The effective diameter of the semicircles decreases with prolonged deposition, which indicates a decrease in the charge transfer resistance of the BDD film. In addition, a decrease in the effective diameter of the semicircles can be seen when the boron flow rate increases with the same deposition time. The enhanced boron doping extent can also reduce the transfer charge resistance of the BDD film. The impedance spectra are fitted with an equivalent circuit using ZsimpWin software. The elements in the equivalent circuit include the electrolyte resistance (R_L_), film resistance (R_film_), charge transfer resistance (R_ct_), constant phase elements (Q_1_ and Q_2_), and diffusional element (Z_w_). Q_1_ and Q_2_ represent the capacitance of the film electrode and the double-layer capacitance, respectively. In the low-frequency region, Z_w_ denotes the Warburg resistance. The fitted values estimated for these four electrodes are presented in Table 1. The quite low relative values (<33.8 × 10^−3^) of the weighted sum of squares (χ^2^) are indicative of the high quality of the fittings. Notable decreases in the film resistance and charge transfer resistance can be noted as the deposition time increases, especially the latter. The electrodes with 0.1B-4 and 0.4B-4 have smaller impedances compared to their counterparts with 0.1B-1 (784.8 vs. 9475 Ω cm^2^) and 0.4B-1 (595.7 vs. 3291 Ω cm^2^). This finding indicates that the increase of film thickness can improve the conductivity and charge transfer rate of the films, consistent with the results of CV.

Fe(CN)_6_^−3/−4^ is a commonly-used redox system to assess an electrode’s electrochemical activity. Compared to other outer-sphere systems, Fe(CN)_6_^−3/−4^ redox system should not be considered outer-sphere due to its abnormal response to carbon electrodes [38,39]. For BDD, several factors are found to strongly influence the electrode kinetics of Fe(CN)_6_^−3/−4^. First of all, the electrochemically active region of polycrystalline BDD film does not continuously cover but discretely distributes over the whole surface, which is affected by the boron doping level. The number of active sites on BDD film electrodes bearing a slight boron doping is smaller than that on the counterpart with heavy boron doping. As a result, a lower electron transfer rate and higher ΔEp are observed for BDD electrodes bearing slight boron doping, as reported by Holt et al. [16]. A slower heterogeneous electron transfer kinetics of Fe(CN)_6_^−3/−4^ can be obtained at the oxygen-terminated BDD surface compared with that at the hydrogen-terminated one [40,41]. Oxygen functional groups are thought to block the adsorption sites, which inhibits the heterogeneous electron transfer of Fe(CN)_6_^−3/−4^ [42,43]. Alternative theories also verify that the negative charge on oxygen-containing groups repels the negatively charged redox species [19]. Additionally, the fraction of exposed edge plane on sp^2^-bonded carbon electrode and the surface cleanliness has also been suggested as the aspects governing the electrode reaction kinetics [44]. However, the sp^2^-bonded carbon content has a negligible effect on the ΔE_p_ or peak current for Fe(CN)_6_^−3/−4^ redox systems [39,45].

The electrochemical measurement results presented above (see Figure 3 and Figure 4) show that the electrode bearing the BDD film with 1-h deposition exhibits a quasi-reversible behavior, while those with 4 h deposition at the same boron flow rate exhibit smaller ΔE_p_, a lower charge transfer resistance, and a higher electron transfer rate. As BDD films were pretreated by a −2 V cathodic polarization in H_2_SO_4_ while maintaining a rich hydrogen atmosphere during the cooling process before the electrochemical test, the electrode surfaces are clean with hydrogen-terminated. It can thus be indicated that the quasi-reversible behavior of electrodes with BDD deposited for 1 h does not originate from the change of the surface chemical state into oxygen termination and surface cleanliness. This electrochemical phenomenon is more similar in the CV curves measured for the BDD film with low boron doping levels, as reported in the literature [9,16,46,47]. Moreover, the results obtained via Raman spectroscopy presented in this paper also prove that the boron doping level in the BDD films with 1 h deposition is less than that of the counterparts with 4 h.

For the BDD films prepared by CVD, the most notable feature lies in a typical increase in the grain size of the growth face and a decrease in grain boundary density with growing the film thickness. In undoped diamond films, the non-diamond phase at grain boundaries (mainly sp^2^ bonds) plays an important role in dictating the electrical conductivity of the films. For BDD films, however, as a p-type dopant, boron can induce the formation of shallow acceptor energy levels, resulting in the electron transition in the valence band. Using secondary ion mass spectrometers and Raman spectrometer, Mort et al. found that boron doping was the main reason for obtaining conductivity instead of the non-diamond phase [48], in consistency with the report of Huang et al. based on the investigation by the auger electron spectroscopy [49]. Although a large number of reports have shown that boron atoms mainly reside in grains [29,30,50], the boron distribution at the grain boundary remains controversial. A. S. Barnard et al. proposed that boron was more likely to be doped at grain boundaries based on density functional theory (DFT) calculations [28]. Dubrovinskaia et al. found that boron was enriched in the intergranular pocket and the boundaries of the diamond synthesized at high temperature and high pressure [51]. Lu et al. employed scanning transmission electron microscopy and spatially resolved electron energy-loss spectroscopy to analyze the distribution of boron in the nano-diamond [30]. Boron was detected to be evenly distributed in the grain, grain boundary, and outer amorphous shell. Nevertheless, boron was not enriched at the grain boundary. However, disputes exist regarding the distribution of boron at grain boundaries. The local electrochemical activity is mainly influenced by the electrical conductivity of the specific crystal surface of a single grain, which is in turn affected by the boron doping, rather than the accumulation of boron or sp^2^ hybrid carbon at grain boundaries [17,52]. Meanwhile, it is found that the conductivity cannot be enhanced at the grain boundary. Even if the ultra-nanocrystalline possesses many sp^2^ carbon regions, its ΔEp is not reduced to 100 mV, similar to other sp^2^-carbon-based electrodes with strong adsorption in the dopamine detection process. However, the ΔEp exceeds 500 mV for dopamine on the ultra-nanocrystalline, similar to the typical high-performance diamond electrode with weak adsorption. It is considered that the effect of grain on electrical conductivity and electrochemical performance is larger than that of grain boundary [39]. Therefore, based on the CVD growth model of polycrystalline diamond, the grain of columnar growth form a larger size crystal face and lower grain boundary density. Consequently, an increased electrochemically active surface area and an enhanced electrochemical activity of the electrode are obtained. Generally, boron-induced changes in Raman spectra reflect that the boron atoms are doped into the diamond lattice. The fixed laser spot size during Raman measurements makes the detectable information regions have less grain boundary as the film thickness increases. Thus, the boron-related Raman signal in the grain is depressed, thus presenting a decrease in the surface doping level. We speculate that the columnar growth mode of polycrystalline diamond should be the main reason for the evolution of boron doping level and electrochemical performance with the variation of the BDD film thickness.

## 3. Materials and Methods

### 3.1. Reagents

Potassium ferricyanide (K_3_[Fe(CN)_6_]) and potassium chloride (KCl) were purchased from Tianjin Recovery Technology Development Co., Ltd. (Tianjin, China). Ultrapure water (≥18 MΩ·cm) was used throughout the experiments. All reagents were of analytical grade and without further purifications.

### 3.2. BDD Film Deposition

A doped silicon sheet (4 × 4 × 1 mm^3^) was selected as the substrate. Before deposition, the ultrasonic cleaning of the doped silicon sheet in alcohol and acetone each for 10 min was carried out. Afterward, ultrasonic seeding in the nano-diamond suspension was performed for the cleaned silicon sheet for 30 min. Ultrasonic oscillation in alcohol for 1 min was the final pretreatment for the silicon sheet. Then, the pretreated samples were placed in a chamber. CH_4_ and H_2,_ with 99.999% purities, were used as the feed gases at flow rates of 1 and 49 sccm, respectively. A mixed gas of B_2_H_6_ and H_2_ (with a volume ratio of 5:95) was adopted as the boron source. The flow rates of B_2_H_6_ and H_2_ were set to 0.1 and 0.4 sccm, respectively. The film deposition was conducted at 700 °C and 3 kPa. The deposition durations varied from 1 to 4 h to regulate the film thickness, and Figure 5 shows the whole deposition process. Four samples were prepared and named 0.1B-1, 0.1B-4, 0.4B-1, and 0.4B-4, where 0.1 and 0.4 represent the flow rate of the boron source, and the number 1 or 4 behind the letter B denotes the deposition time. All of the prepared films are hydrogen terminated under the present conditions. At the end of the deposition, the flows of CH_4_ and B_2_H_6_ ceased. Meanwhile, the chamber was cooled down until room temperature. During the whole cooling process, the films were exposed to an H_2_ atmosphere.

### 3.3. Apparatus

Field emission scanning electron microscopy (Nova Nano SEM 230, FEI Co., Hillsboro, OR, USA) was used to characterize the surface morphology of the prepared BDD thin films. Micro Raman spectrometer (LabRAM HR800, Horiba Jobin Yvon, Paris, France) was used to analyze the composition of these BDD films, where an Ar^+^ laser was adopted as the light source at an excitation wavelength of 488 nm.

### 3.4. Electrochemical Measurements

All electrochemical experiments were carried out with the electrochemical workstation (Shanghai Chenhua CHI660E, Shanghai, China). A three-electrode system was used for the electrochemical experiments. Saturated Ag/AgCl, a platinum sheet (1 × 1 cm^2^), and the BDD film were employed as the reference, auxiliary, and working electrodes, respectively. For EIS measurements, the working electrode was tested at an open circuit potential over the frequency range of 10^−2^–10^6^ Hz with an amplitude of 5 mV. All the prepared electrodes were pretreated at an applied potential of −2 V in a 1 M H_2_SO_4_ solution for 2 min.

## 4. Conclusions

BDD films with different thicknesses were prepared by HFCVD. Adjusting the deposition time allowed the fine-tuning of the film thickness. The influence of the BDD film thickness on the boron doping level and electrochemical properties was investigated. The SEM observation revealed that the grain size and boundary density were increased and lowered with an increase in the film thickness, respectively. The intensity and the asymmetry degree of the Raman scattering band at 1332 cm^−1^ were enhanced with increasing the film thickness, indicating that the thickness increase promoted the boron doping. The results of CV and EIS in a K_3_[Fe(CN)_6_] solution showed a smaller peak potential difference, a larger peak current, and a lower charge transfer resistance for the electrodes with the BDD film prepared by longer-time deposition. It was proved that the active sites on the BDD surface could be enriched by longer-time deposition. We suggested that this was mainly due to the columnar crystal growth mode of the polycrystalline diamond film. An increase in the film thickness enabled the growth of the grains and a decrease in the sp^2^ carbon at the grain boundaries. Consequently, more boron atoms were incorporated into the diamond lattice.

## Figures and Tables

**Figure 1 molecules-28-02829-f001:**
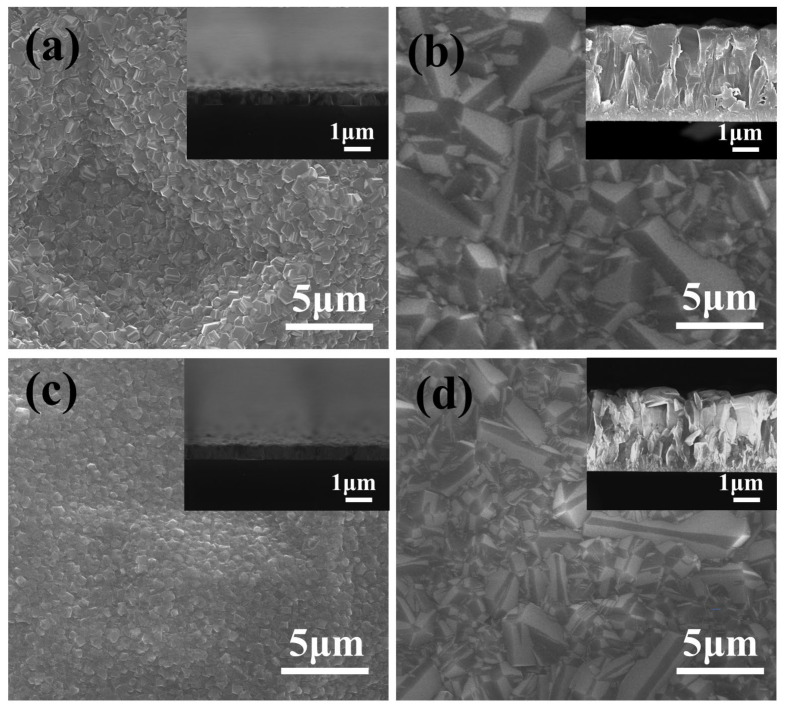
SEM images of BDD films with various deposition time and boron doping level: (**a**) 1 h with 0.1 B_2_H_6_; (**b**) 4 h with 0.1 B_2_H_6_; (**c**) 1 h with 0.4 B_2_H_6_; (**d**) 4 h with 0.4 B_2_H_6_. The inset is the corresponding cross-section.

**Figure 2 molecules-28-02829-f002:**
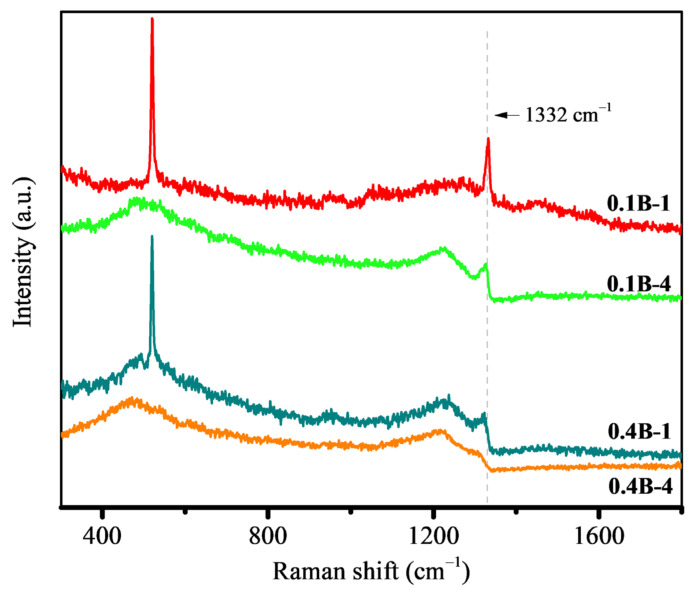
Raman spectra of different BDD films.

**Figure 3 molecules-28-02829-f003:**
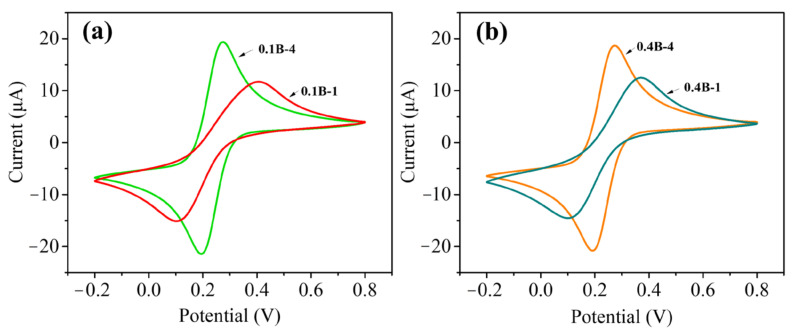
CVs for different BDD electrodes in 1 mM K_3_[Fe(CN)_6_] with 0.1 M KCl at a scan rate of 50 mv/s: (**a**) 1 h with 0.1 B_2_H_6_ and 4 h with 0.1 B_2_H_6_; (**b**) 1 h with 0.4 B_2_H_6_ and 4 h with 0.4 B_2_H_6_.

**Figure 4 molecules-28-02829-f004:**
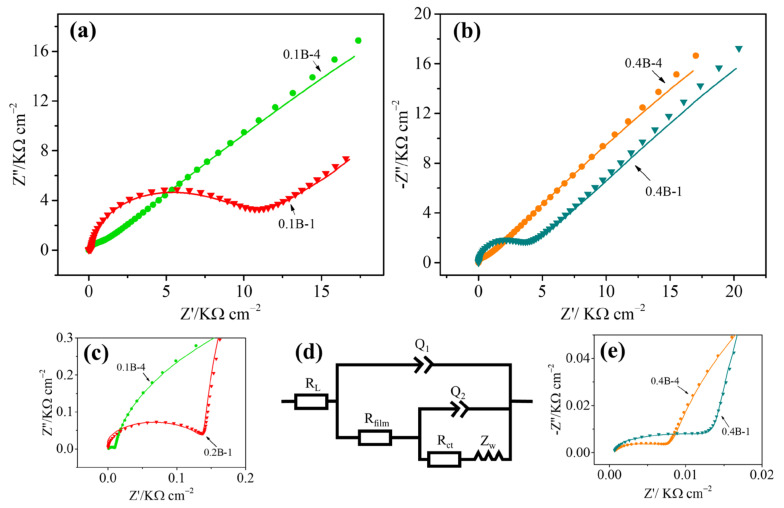
(**a**,**b**) Impedance spectra of different BDD electrodes in 1mM K_3_[Fe(CN)_6_] with 0.1 M KCl electrolyte, (**c**,**e**) Enlarged impedance spectra of the BDD electrodes with 0.1 B_2_H_6_ and 0.4 B_2_H_6_ doping, respectively. (**d**) the Equivalent circuit.

**Figure 5 molecules-28-02829-f005:**
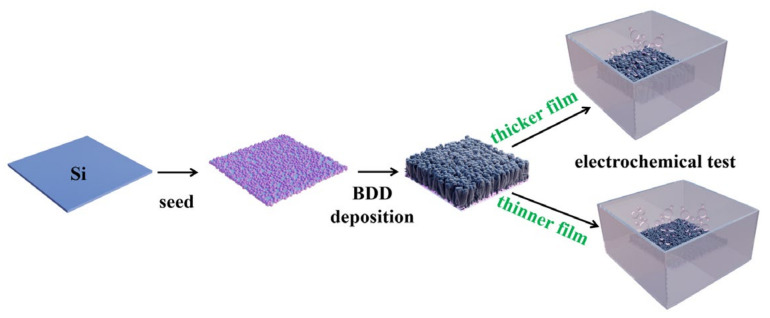
Schematic diagram of the BDD electrode preparation process.

**Table 1 molecules-28-02829-t001:** Electrochemical parameters obtained based on EIS tests on different electrodes in a mixed solution of 1mM K_3_[Fe(CN)_6_] and 0.1M KCl.

Sample	R_L_ (Ω cm^2^)	Q_1_ (μF cm^−2^ s^n−1^)	n_1_	R_film_ Ω cm^2^	Q_2_ (μF cm^−2^ s^n−1^)	n_2_	R_ct_ Ω cm^2^	Z_W_ (mF cm^−2^ s)	χ^2^ (10^−4^)
0.1B-1	0.68	0.142	1	16.6	2.56	0.969	9475	0.165	33.81
0.1B-4	0.701	0.173	0.98	9.98	6.71	0.939	784.8	0.167	7.741
0.4B-1	0.64	0.215	1	15.9	2.74	0.961	3291	0.164	13.45
0.4B-4	0.617	0.237	0.99	8.07	6.13	0.938	595.7	0.169	7.745

## Data Availability

Data are available with requirements.

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
