# Peer review of "Thickness Effects on Boron Doping and Electrochemical Properties of Boron-Doped Diamond Film"

_molecules, 2023, doi:10.3390/molecules28062829_

Round 1
Reviewer 1 Report
The article „Thickness effects on boron doping and electrochemical properties of boron doped diamond film“ from authors Hangyu Long et al. actually is a well-written manuscript with a broad and complex introduction, clear and complex interpretation of results supported by SEM, Raman, CV, EIS measurements. The description of experimental part is also OK, and the references seems to be also good, maybe some new works good be added because there are no references from last 6 years.
However, the main problem of the article is in its concept. The article is about the influence of the thickness on boron doping of BDD electrodes, but there is no information about the thickness of BDD and no information about the boron concentration in the analysed BDD films. I.e. what is the thickness of the BDD film deposited for 1 hour and 4 hours, and what is the boron level in these films? These are key parameters for the designed concept of the article. There is only one sentence (page 4, line 117), that based on Raman measurements the level of boron dopation should be >2×1020 cm-3 : „The existence of this asymmetry means that the doping level of boron exceeds 2×1020 cm-3 and is considered metal-like[31,32]“. This consideration is not very precise, and additional technique is needed to determine the boron concentration of the studied BDD films, and the thickness of the BDD films should be also added. Without these information (thickness and boron doping of individual BDD film) the article does not make sense.
Some additional comments:
- Page 3: Changing the contrast in Fig. 1 could improve the quality of the images
- Page 4: in Fig. 3 there is a scan rate 50 mv/s while in the text 100 mv/s (line 144)… need to be corrected, and of course use „mV“ instead of „mv“
- Page 7, line 266 „mV“ instead of „mv“
- Page 7, line 273: „The grain boundaries of polycrystalline diamond have a ~10 nm thick amorphous carbon“ – how did you measure it, where does this information come from?
Reviewer 2 Report
The 2. Results and Discussions are exchanged with the 3. Materials and Methods
What is the innovation factor of this article when compared to the literature?
The manuscript says“...It can be seen that no obvious characteristic band at about 1350 cm-1 and 1500 cm-1 104 corresponding to non-diamond was detectable, indicating that the quality of the film is 105 good...” The Raman spectrum 0.1B-1 shows a small band at 1350 cm-1.
When the level of Boron doping in the diamond films is increased, an extensive band appears at 1220 cm-1 n due to the presence of a very high Boron concentration in the diamond lattice. However, the band at 1220 cm-1 spectra of 0.1B-4 and 0.4B-1 and 0.4B-1 are similar.
How thick are the diamond films?
Round 2
Reviewer 1 Report
no additional comments
Reviewer 2 Report
The manuscript quality has been increased.